# Spik-NeRF:
# Spiking Neural Networks for Neural Radiance Fields

**Gang Wan**[1,*]**, Qinlong Lan**[1,*]**, Zihan Li**[2,3]**, Huimin Wang**[2,3]**,**
**Yitian Wu**[1]**, Zhen Wang**[1]**, Wanhua Li**[1]**, Yufei Guo**[3,†]
[1]Space Engineering University, [2]Peking University
[3]Intelligent Science & Technology Academy of CASIC
casper_51@163.com, whulanql@whu.edu.cn, yfguo@pku.edu.cn

## Abstract

Spiking Neural Networks (SNNs), as a biologically inspired neural network archi-
tecture, have garnered significant attention due to their exceptional energy efficiency
and increasing potential for various applications. In this work, we extend the use
of SNNs to neural rendering tasks and introduce Spik-NeRF (Spiking Neural Radi-
ance Fields with Ternary Spike). We observe that the binary spike activation map of
traditional SNNs lacks sufficient information capacity, leading to information loss
and a subsequent decline in the performance of spiking neural rendering models.
To address this limitation, we propose the use of ternary spike neurons, which
enhance the information-carrying capacity in the spiking neural rendering model.
With ternary spike neurons, Spik-NeRF achieves performance that is on par with,
or nearly identical to, traditional ANN-based rendering models. Additionally, we
present a re-parameterization technique for inference that allows Spik-NeRF with
ternary spike neurons to retain the event-driven, multiplication-free advantages
typical of binary spike neurons. Furthermore, to further boost the performance
of Spik-NeRF, we employ a distillation method, using an ANN-based NeRF to
guide the training of our Spik-NeRF model, which is more compatible with the
our ternary neurons compared to the standard binary neurons and other neuron
forms. We evaluate Spik-NeRF on both realistic and synthetic scenes, and the
experimental results demonstrate that Spik-NeRF achieves rendering performance
comparable to ANN-based NeRF models.

## 1 Introduction

Spiking Neural Networks (SNNs) [36, 3, 9, 10, 35, 18, 17], known for their energy efficiency
compared to Artificial Neural Networks (ANNs), have garnered significant attention due to their event-
driven computation mechanism and the energy-saving advantages of multiplication-free operations.
SNNs have shown great potential in a wide range of applications. For instance, in [34], SNNs
were applied to object detection and demonstrated substantial energy efficiency improvements,
outperforming their ANN counterparts by orders of magnitude. In [16], SNNs were used to improve
the image de-occlusion task. In [33], SNNs were employed for sequential learning, showing better
performance and reduced energy consumption compared to ANNs with similar scale. Similarly,
in [26], SNNs were utilized for Human Activity Recognition (HAR), achieving up to a 94% reduction
in energy consumption while maintaining comparable accuracy to ANN-based models. Additionally,
SNNs have been applied to pose tracking [45], 3D recognition [37], and even autonomous driving [39],

---

*Equal Contributions.
†Corresponding Author.

39th Conference on Neural Information Processing Systems (NeurIPS 2025).

where LaneSNNs demonstrated low power consumption ( 1 W) in lane detection using event-based cameras.

Given these successes, the question naturally arises: *Can SNNs be adapted to the more complex task of neural rendering, such as rendering neural radiance fields (NeRF)?*

In this paper, we introduce **Spik-NeRF**, a spiking neural network approach tailored for neural rendering tasks, the first one directly-trained SNN-based NeRF model building upon the initial NeRF framework [31]. However, we found that applying SNNs to neural rendering tasks led to suboptimal performance. This is primarily due to the limited information capacity of the binary spike activation maps in SNNs. Unlike the rich activation maps of ANNs, the binary spike maps in SNNs fail to retain enough useful information during the quantization process, resulting in significant information loss and a decrease in performance. A more detailed explanation is provided in Sec. 3.3.

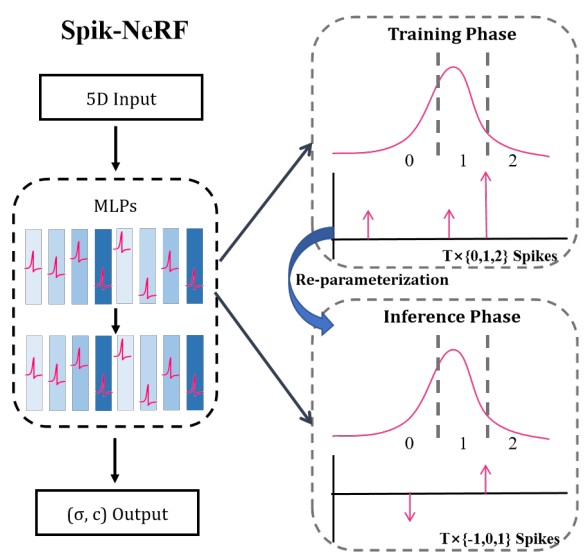

Figure 1: The overall workflow of the proposed Spik-NeRF, along with the ternary spike neuron and re-parameterization technique.

To address this challenge, we propose the ternary spike neuron for the Spik-NeRF, which extends the traditional binary spike representation ($\{0, 1\}$) to a ternary form ($\{0, 1, 2\}$). This new approach significantly increases the information capacity of SNNs, as detailed in Sec. 3.3. Furthermore, in the inference phase, we introduce a re-parameterization technique that transforms the ternary spikes ($\{0, 1, 2\}$) into the set of ($\{-1, 0, 1\}$), preserving the multiplication-free and event-driven advantages of SNNs. The overall workflow of the proposed Spik-NeRF, along with the ternary spike neuron and re-parameterization technique, is illustrated in Fig. 1.

Additionally, to further improve the performance of Spik-NeRF, we propose using an ANN-based NeRF model for distillation. This technique, leveraging the superior capabilities of ANN-based NeRF models, is particularly well-suited for our ternary spike neuron, offering additional performance gains.

In summary, the main contributions of this work are as follows:

- We present Spik-NeRF, a spiking neural network for rendering neural radiance fields. To our best knowledge, this is the first directly-trained SNN model building on the original NeRF framework [31].

- We demonstrate, with theoretical justification, that binary spike activation maps in SNN-based NeRF are insufficient in carrying information, leading to performance degradation. To solve this issue, we propose the ternary spike neuron, which effectively increases the information capacity while retaining the multiplication-free and event-driven advantages of standard SNNs in the inference, aided by our re-parameterization technique.

- We introduce a distillation method using an ANN-based NeRF teacher, which is more suitable for our ternary neuron compared to other spike neurons, to further enhances the performance of Spik-NeRF.

- We evaluate Spik-NeRF on both realistic and synthetic scenes. The experimental results demonstrate that Spik-NeRF achieves rendering performance comparable to ANN-based NeRF models.

## 2 Related Work

### 2.1 Spiking Neural Networks

There are generally three primary methods for training SNNs [13]: (1) spike-timing-dependent plasticity (STDP) [1] approaches, (2) ANN-to-SNN conversion approaches [22, 21, 29, 6, 8, 2, 19, 24], and (3) direct training methods [5, 32, 42, 35, 25, 41, 40].

The STDP method is biologically inspired [20, 7] and updates synaptic weights using an unsupervised learning algorithm called Hebbian learning [23]. However, this approach is still limited to small-scale datasets.

The ANN-to-SNN conversion method [6, 24, 22, 21] involves converting a well-trained ANN model to an SNN counterpart. This method is advantageous because training an ANN is faster than training an SNN. Consequently, the ANN-to-SNN conversion offers a quick way to obtain an SNN without using gradient descent. However, the converted SNN essentially mimics the original ANN and lacks learned features, thus not fully exploiting the benefits of SNNs. Additionally, this method typically requires many time steps to achieve high accuracy.

Direct training methods, on the other hand, aim to find an alternative function to replace the firing function of spiking neurons during backpropagation. These methods can significantly reduce the number of time steps needed, sometimes even to fewer than five [12, 25, 14], and have attracted considerable attention recently. However, finding an appropriate surrogate function for SNNs with large time steps remains a challenging problem. Our work focuses on addressing this issue.

### 2.2 Spiking Neural Networks for NeRF

Some research has explored applying SNNs to Neural Radiance Fields, but these studies differ from our approach. For instance, hybrid ANN-SNN models were proposed in Spiking NeRF [28] and Spiking Nerfacto [11]. These works employ non-linear, non-spike functions to post-process the density-related outputs of the original ANN-based NeRF models. In contrast, our work focuses on utilizing a fully SNN-based model for neural radiance fields. In Spiking-Nerf [27], an ANN-SNN model is proposed to develop energy-efficient spiking neural rendering using the ANN-to-SNN conversion approach, which, as mentioned earlier, requires many time steps. In comparison, our work concentrates on developing a direct training SNN method for NeRF. Another relevant study, SpikingNerf [43], presents a spiking neural radiance field model based on the DVGO [38] and TensoRF [4] frameworks. These frameworks enhance the original NeRF model by integrating various innovations. Except for these, this method requires numerous time steps as each sampled point on the ray is associated with a particular time step and represented in a hybrid manner.

Our approach, however, aims to develop a spiking neural radiance field model based on the original NeRF framework [31], which could pave the way for future advancements in this field. Additionally, we aim to achieve competitive performance with fewer time steps, which is directly related to the energy efficiency of the model.

## 3 Preliminary and Methodology

In this paper, we primarily apply the SNN to the NeRF framework [31], which is the first deep learning model that represents a scene as a neural radiance field and renders novel views from this representation. We then modify it to create Spik-NeRF. First, we provide a detailed introduction to NeRF and the widely used SNN neuron model, the Leaky Integrate-and-Fire (LIF) model. Subsequently, we address the information loss issue when applying SNN to NeRF. Finally, we present the Spik-NeRF model, which is based on ternary spike neurons to resolve the aforementioned problem, along with an isomorphic network knowledge distillation method to further enhance performance.

### 3.1 NeRF

In contrast to conventional explicit 3D reconstruction techniques that employ discrete voxel grids or point clouds, NeRF [31] introduces a novel continuous implicit representation through differentiable volumetric rendering. The core innovation lies in encoding the 3D scene as a *5D neural radiance*

*field* using a multi-layer perceptron (MLP), which maps spatial coordinates $\mathbf{p} = (x, y, z) \in \mathbb{R}^3$ and viewing directions $\mathbf{v} = (\theta, \phi) \in \mathbb{S}^2$ to volume density $\sigma \in \mathbb{R}^+$ and view-dependent RGB color $\mathbf{c} = (r, g, b) \in [0, 1]^3$. This parametric representation is formalized through two cascaded MLP components:

$$\text{Geometry network:} \quad F_\theta : \mathbf{p} \mapsto (\mathbf{e}, \sigma) \tag{1}$$

$$\text{Appearance network:} \quad G_\gamma : (\mathbf{e}, \mathbf{v}) \mapsto \mathbf{c} \tag{2}$$

where $\theta$ and $\gamma$ denote network parameters, $\mathbf{e} \in \mathbb{R}^N$ represents intermediate feature embeddings, and $\sigma$ corresponds to the differential opacity at point $\mathbf{p}$. These networks are all composed of several fully connected layers. Each fully connected layer implements the transformation:

$$\mathbf{h} = \text{ReLU}(\text{Linear}(\mathbf{a})) \tag{3}$$

where $\mathbf{a}$ is the activation from the previous layer.

The rendering process employs classical volume rendering principles [30] with neural adaptation. For a camera ray $\mathbf{r}(g) = \mathbf{o} + g\mathbf{d}$ with near/far bounds $[g_n, g_f]$, we sample $K$ stratified points $\{g_i\}_{i=1}^K$ and compute the pixel color via numerical integration:

$$\hat{C}(\mathbf{r}) = \sum_{i=1}^K G_i \alpha_i \mathbf{c}_i \quad \text{where} \quad \begin{cases} G_i = \exp\left(-\sum_{j=1}^{i-1} \sigma_j \delta_j\right) \\ \alpha_i = 1 - \exp(-\sigma_i \delta_i) \\ \delta_i = g_{i+1} - g_i \end{cases} \tag{4}$$

Here $G_i$ represents the transmittance probability for ray segment $[g_n, g_i]$, and $\alpha_i$ denotes the alpha-compositing weight for the $i$-th sample. The hierarchical sampling strategy combines coarse and fine networks to importance-sample along rays.

The model is optimized through photometric reconstruction over a set of rays $\mathcal{R}$ using an $L_2$ loss between rendered and observed pixel colors:

$$\mathcal{L}_{\text{photo}} = \mathbb{E}_{\mathbf{r} \sim \mathcal{R}} \left[ \|\hat{C}_c(\mathbf{r}) - C(\mathbf{r})\|_2^2 + \|\hat{C}_f(\mathbf{r}) - C(\mathbf{r})\|_2^2 \right] \tag{5}$$

where $\hat{C}_c$ and $\hat{C}_f$ denote outputs from the coarse and fine networks respectively.

## 3.2 Vanilla SNN for NeRF (Denoted as Spiking NeRF)

SNNs use the spiking neuron, which is inspired by the brain's natural mechanisms, to transmit information. A spiking neuron will receive input spike trains from the previous layer neuron models along times to update its membrane potential, $\mathbf{u}$. In the paper, we adopt the widely used leaky integrate and fire (LIF) neuron model. The LIF neuron model governs membrane potential $\mathbf{u}(t)$ evolution through time $t$:

$$\tau_{\text{m}} \frac{d\mathbf{u}}{dt} = -(\mathbf{u} - u_{\text{reset}}) + R \cdot I(t), \quad \mathbf{u} < V_{\text{th}} \tag{6}$$

$$\mathbf{s}^t = \Theta(\mathbf{u}^t - V_{\text{th}}) \tag{7}$$

where $\Theta(\cdot)$ denotes the Heaviside step function, $\tau_{\text{m}}$ is the membrane time constant, and $V_{\text{th}}$ the firing threshold. For practical implementation, we adopt the discrete-time formulation:

$$\mathbf{v}^t = \mathbf{u}^{t-1} + \frac{1}{\tau}(\mathbf{W}\mathbf{s}^{t-1} - \mathbf{u}^{t-1} + u_{\text{reset}}) \tag{8}$$

$$\mathbf{s}^t = \Theta(\mathbf{v}^t - V_{\text{th}}) \tag{9}$$

$$\mathbf{u}^t = \mathbf{v}^t \odot (1 - \mathbf{s}^t) + u_{\text{reset}}\mathbf{s}^t \tag{10}$$

where $\mathbf{W}$ denotes synaptic weights and $\odot$ the Hadamard product. Parameters follow biological constraints: $\tau = 4$, $u_{\text{reset}} = 0$, $V_{\text{th}} = 0.5$ [25].

We transform NeRF's MLP layers to spiking domains through temporal unfolding and potential accumulation:

$$\text{Spiking Geometry Network:} \quad F_\theta^t : \mathbf{p}^t \mapsto (\mathbf{e}^t, \sigma^t) \tag{11}$$

$$\text{Spiking Appearance Network:} \quad G_\gamma^t : (\mathbf{e}^t, \mathbf{v}^t) \mapsto \mathbf{c}^t \tag{12}$$

Each spiking MLP layer implements temporal-aware computation:

$$\mathbf{h}_k^t = \text{LIF}(\mathbf{W}_k \mathbf{s}_{k-1}^t + \mathbf{b}_k) \tag{13}$$

where $\mathbf{s}_{k-1}^t$ denotes spike inputs from layer $k-1$ at timestep $t$. The membrane potential $\mathbf{u}_k^t$ tracks temporal dependencies across layers.

The spiking radiance field outputs are integrated through:

$$\bar{\sigma} = \frac{1}{T} \sum_{t=1}^{T} \sigma^t \tag{14}$$

$$\bar{\mathbf{c}} = \frac{1}{T} \sum_{t=1}^{T} \mathbf{c}^t \tag{15}$$

where $T$ is the total timestep of the Spiking NeRF. Then volumetric rendering then follows Eq. (4) with $\sigma_i = \bar{\sigma}_i$ and $\mathbf{c}_i = \bar{\mathbf{c}}_i$.

### 3.3 Spik-NeRF

#### 3.3.1 Information Loss in Spiking NeRF

While employing binary spike feature embeddings in Spiking NeRF offers substantial energy efficiency, it inherently has a limited representational capacity compared to the high-precision feature embeddings utilized in ANN-based NeRF. This limitation ultimately restricts its performance. To better illustrate this issue, we begin by providing a theoretical analysis based on the concept of entropy. Given a set $\mathbf{X}$, its representational capability, denoted as $\mathcal{C}(\mathbf{X})$, can be quantified by the maximum entropy of $\mathbf{X}$, as expressed below:

$$\mathcal{C}(\mathbf{X}) = \max \mathcal{H}(\mathbf{X}) = \max \left( -\sum_{x \in \mathbf{X}} p_{\mathbf{X}}(x) \log p_{\mathbf{X}}(x) \right), \tag{16}$$

where $p_{\mathbf{X}}(x)$ represents the probability of observing a sample $x$ from $\mathbf{X}$. The following proposition can be easily derived:

**Proposition:** *Given a set X, we have $\mathcal{C}(X) = \max \mathcal{H}(X) = \max \left( -\sum_{x \in X} p_X(x) \log p_X(x) \right)$. When the probability distribution is defined as $p_X(x) = \frac{1}{M}$, where $M$ represents the total number of samples in X, the entropy $\mathcal{H}(X)$ reaches its maximum value of $\log(M)$. Therefore, it follows that $\mathcal{C}(X) = \log(M)$.*

Next, we calculate the representational capacities of the binary spike feature embeddings in Spiking NeRF and the real-valued feature embeddings in the ANN-based counterpart. Let $\mathbf{E}_{\text{B}} \in \mathbb{B}^{C \times N}$ represent the binary feature embeddings of the Spiking NeRF, and $\mathbf{E}_{\text{R}} \in \mathbb{R}^{C \times N}$ denote the real-valued feature embeddings of the ANN-based NeRF. A binary spike output $s$ can be represented by 1 bit, with two possible samples from $s$. Therefore, the number of samples in $\mathbf{E}_{\text{B}}$ is $2^{(C \times N)}$, and the corresponding representational capacity is:

$$\mathcal{C}(\mathbf{E}_{\text{B}}) = \log \left( 2^{(C \times N)} \right) = C \times N. \tag{17}$$

In contrast, a real-valued output requires 32 bits, leading to $2^{32}$ possible samples. Hence, the representational capacity for the real-valued embeddings is:

$$\mathcal{C}(\mathbf{E}_{\text{R}}) = \log \left( 2^{32 \times (C \times N)} \right) = 32 \times C \times N. \tag{18}$$

This comparison clearly demonstrates that the representational capacity of the binary spike feature embeddings is substantially limited, which consequently results in degraded performance for Spiking NeRF.

#### 3.3.2 Ternary Spike Neuron Mechanism for Spik-NeRF

Our theoretical analysis reveals that enhancing the information capacity of spike neuron activations directly correlates with improved task performance. To capitalize on this insight, we propose a novel

ternary spike neuron formulation that forms the foundation of our Spik-NeRF architecture. The membrane dynamics and spike generation mechanism operate through three distinct phases:

$$\mathbf{v}^t = \mathbf{u}^{t-1} + \frac{1}{\tau}\left(\mathbf{W}\mathbf{s}^{t-1} - \mathbf{u}^{t-1} + u_{\text{reset}}\right) \tag{19}$$

$$\mathbf{s}^t = \begin{cases} 2, & \mathbf{v}^t > V_{\text{th}} + \Delta_v \\ 1, & V_{\text{th}} \leq \mathbf{v}^t \leq V_{\text{th}} + \Delta_v \\ 0, & \mathbf{v}^t < V_{\text{th}} \end{cases} \tag{20}$$

$$\mathbf{u}^t = \begin{cases} \mathbf{v}^t, & \mathbf{v}^t < V_{\text{th}} \\ u_{\text{reset}}, & \text{otherwise} \end{cases} \tag{21}$$

where $\Delta_v$ represents our adaptive threshold margin (fixed at 1 in implementation). Obviously, this ternary formulation significantly enhances the representational capacity of Spik-NeRF. To quantitatively analyze the representational advantage, we also resort to the information entropy theory. Let $\mathbf{E}_T \in \mathbb{T}^{C \times N}$ denote as a ternary feature embedding in our Spik-NeRF. The ternary spike embeddings $\mathbf{E}_T$ consists of $3^{C \times N}$ samples. Hence,

$$\mathcal{C}(\mathbf{E}_T) = \log_2 3^{C \times N} = C \times N \cdot \log_2 3 \approx 1.585 \cdot C \times N \tag{22}$$

The $\approx 58.5\%$ increase in theoretical information capacity directly translates to enhanced scene representation capabilities, which benefits performance improvement.

### 3.3.3 Training-Inference Decoupling via Spike Reparameterization

While ternary spike neurons enhance representational capacity, their direct implementation introduces computational challenges: the $\{0, 1, 2\}$ activation space prevents efficient conversion of weight-activation multiplications to additions, a critical advantage in SNN acceleration. To resolve this fundamental efficiency conflict, we propose a novel spike re-parameterization technique that preserves both the information richness of ternary signals and the computational benefits of binary networks.

Our solution employs a train-infer decoupling strategy with affine transformation of spike representations. During training, we maintain the native $\{0, 1, 2\}$ spike formulation for gradient stability. For inference, we apply a linear transformation to the spike tensor:

$$\hat{\mathbf{s}}^t = \mathbf{s}^t - \Delta \cdot \mathbf{1} \quad \text{where} \quad \Delta = 1 \tag{23}$$

This shifts the spike space to $\{-1, 0, 1\}$ while preserving ordinal relationships. The membrane potential update equations consequently adapt as follows:

$$\text{Training:} \quad \mathbf{v}^t = \mathbf{u}^{t-1} + \frac{1}{\tau}\left(\mathbf{W}\mathbf{s}^{t-1} - \mathbf{u}^{t-1} + u_{\text{reset}}\right) \tag{24}$$

$$\text{Inference:} \quad \mathbf{v}^t = \mathbf{u}^{t-1} + \frac{1}{\tau}\left(\mathbf{W}\hat{\mathbf{s}}^{t-1} + \mathbf{w}_b - \mathbf{u}^{t-1} + u_{\text{reset}}\right) \tag{25}$$

where $\mathbf{w}_b = \mathbf{W}\mathbf{1}$ constitutes a pre-computable bias term. Thus in the inference, the linear layer will only consist of addition operations and keep the event-driven advantage.

Note that we can also use the $\{-1, 0, 1\}$ activation spike [15] during training. However, we observe that its performance is inferior to that of the $\{0, 1, 2\}$ spike. Additionally, the $\{0, 1, 2\}$ spike activation resembles ReLU activation more closely, making it better suited for ANN-SNN distillation compared to the $\{-1, 0, 1\}$ spike activation. With these two reasons, we chose this form of neuron for our work.

### 3.3.4 Isomorphic Network Knowledge Distillation

To further increase the performance, we propose an isomorphic distillation framework that transfers knowledge from an ANN-based NeRF (teacher) to our Spik-NeRF (student). We establish direct supervision on both density and color predictions through mean squared error (MSE) distillation. For any 3D point $\mathbf{p}$ and viewing direction $\mathbf{v}$,

$$\mathcal{L}_{\text{density}} = \mathbb{E}_{\mathbf{p} \sim \mathcal{P}}\left[\|\sigma^{\text{ANN}}(\mathbf{p}) - \bar{\sigma}^{\text{SNN}}(\mathbf{p})\|_2^2\right] \tag{26}$$

$$\mathcal{L}_{\text{color}} = \mathbb{E}_{(\mathbf{p},\mathbf{v}) \sim \mathcal{P} \times \mathcal{V}}\left[\|\mathbf{c}^{\text{ANN}}(\mathbf{p}, \mathbf{v}) - \bar{\mathbf{c}}^{\text{SNN}}(\mathbf{p}, \mathbf{v})\|_2^2\right] \tag{27}$$

The final training objective combines photometric reconstruction loss with distillation is

$$\mathcal{L}_{\text{total}} = \mathcal{L}_{\text{photo}} + \lambda_d \mathcal{L}_{\text{density}} + \lambda_c \mathcal{L}_{\text{color}} \tag{28}$$

where $\lambda_d$ and $\lambda_c$ control the distillation strength for density and color respectively. We set them as 0.5 in the paper.

Table 1: Per-scene quantitative results from the synthetic dataset

| Metric | Method | Chair | Drums | Ficus | Hotdog | Lego | Materials | Mic | Ship | Avg. |
|--------|--------|-------|-------|-------|--------|------|-----------|-----|------|------|
| PSNR↑ | ANN-based NeRF | 34.15 | 25.64 | 29.15 | 36.85 | 31.48 | 29.34 | 33.12 | 29.42 | 31.15 |
| | Spiking-NeRF | 12.24 | 11.14 | 13.98 | 14.85 | 9.81 | 10.36 | 9.81 | 13.22 | 11.93 |
| | Spiking NeRF | 31.98 | 24.51 | 24.00 | 34.55 | 29.33 | 27.75 | 31.65 | 27.85 | 28.95 |
| | Spik-NeRF | 33.40 | 25.21 | 26.05 | 35.99 | 30.82 | 28.86 | 32.78 | 28.75 | 30.23 |
| SSIM↑ | ANN-based NeRF | 0.979 | 0.929 | 0.966 | 0.979 | 0.965 | 0.958 | 0.978 | 0.874 | 0.953 |
| | Spiking NeRF | 0.963 | 0.907 | 0.891 | 0.969 | 0.940 | 0.937 | 0.969 | 0.838 | 0.927 |
| | Spik-NeRF | 0.973 | 0.921 | 0.929 | 0.976 | 0.957 | 0.950 | 0.975 | 0.856 | 0.942 |
| LPIPS↓ | ANN-based NeRF | 0.014 | 0.053 | 0.022 | 0.015 | 0.020 | 0.024 | 0.023 | 0.086 | 0.032 |
| | Spiking NeRF | 0.034 | 0.086 | 0.119 | 0.032 | 0.039 | 0.047 | 0.043 | 0.126 | 0.066 |
| | Spik-NeRF | 0.021 | 0.065 | 0.063 | 0.021 | 0.027 | 0.033 | 0.028 | 0.105 | 0.045 |

Table 2: Per-scene quantitative results from the realistic dataset

| Metric | Method | Room | Fern | Leaves | Fortress | Orchids | Flower | T-Rex | Horns | Avg. |
|--------|--------|------|------|--------|----------|---------|--------|-------|-------|------|
| PSNR↑ | ANN-based NeRF | 31.38 | 26.25 | 21.98 | 31.35 | 21.20 | 27.51 | 27.27 | 28.10 | 26.88 |
| | Spiking-NeRF | 17.07 | 15.91 | 9.68 | 14.51 | 9.12 | 10.35 | 15.05 | 13.05 | 13.09 |
| | Spiking NeRF | 30.12 | 25.08 | 20.75 | 30.07 | 20.51 | 26.19 | 25.26 | 26.34 | 25.54 |
| | Spik-NeRF | 30.90 | 25.70 | 21.46 | 30.79 | 20.94 | 26.94 | 26.15 | 27.16 | 26.26 |
| SSIM↑ | ANN-based NeRF | 0.931 | 0.836 | 0.790 | 0.896 | 0.734 | 0.853 | 0.896 | 0.877 | 0.851 |
| | Spiking NeRF | 0.904 | 0.769 | 0.691 | 0.837 | 0.644 | 0.782 | 0.825 | 0.800 | 0.781 |
| | Spik-NeRF | 0.920 | 0.802 | 0.744 | 0.869 | 0.688 | 0.820 | 0.862 | 0.835 | 0.817 |
| LPIPS↓ | ANN-based NeRF | 0.049 | 0.101 | 0.119 | 0.059 | 0.122 | 0.075 | 0.062 | 0.078 | 0.083 |
| | Spiking NeRF | 0.098 | 0.207 | 0.203 | 0.130 | 0.223 | 0.134 | 0.135 | 0.166 | 0.162 |
| | Spik-NeRF | 0.066 | 0.164 | 0.157 | 0.092 | 0.170 | 0.101 | 0.095 | 0.123 | 0.121 |

# 4 Experiment

We assess the rendering performance of Spik-NeRF on both synthetic and real-world datasets [31]. The synthetic dataset includes eight scenes featuring different objects. For each scene, there are 100 views used for training and 200 views for testing, with each view image having a resolution of $400 \times 400$ pixels. The real-world dataset consists of eight scenes captured with mobile phones. Each scene contains between 20 and 60 images, and the images are resized to $400 \times 400$ pixels in this paper. Additionally, one-eighth of the images are reserved for testing.

The network architecture follows the design outlined in NeRF [31]. All models are trained using the Adam optimizer for 300,000 iterations with a batch size of 1,024 rays. We initialize the learning rate at $5 \times 10^{-4}$, which is decayed exponentially as training progresses. For synthetic scenes, the number of sampled points is set to 64 for the coarse network and 128 for the fine network. Similarly, for real-world scenes, 64 and 128 sampled points are used for the coarse and fine networks, respectively. The total number of timesteps for both Spiking NeRF and our Spik-NeRF is set to 2, while for the Spiking-NeRF, it is 8 timesteps. Since our Spik-NeRF with 2 timesteps achieves rendering performance comparable to the ANN-based NeRF model, we did not explore larger timesteps.

## 4.1 Rendering Performance

We evaluate the rendering performance of Spik-NeRF both quantitatively and qualitatively. Tables 1 and 2 present per-scene quantitative results from the synthetic and realistic datasets, respectively.

As mentioned earlier, although previous work has explored the application of SNNs to NeRF, these studies differ from our approach. Spiking-NeRF [27], an ANN-SNN hybrid model, uses the same NeRF [31] framework as ours, and is thus selected for comparison. Additionally, we implemented the original NeRF [31] and a spiking NeRF based on a vanilla binary spike neuron for comparison, referred to as ANN-based NeRF and Spiking NeRF, respectively.

For performance evaluation, we adopt standard metrics: PSNR and SSIM (higher values are better), and LPIPS [44] (lower values are better), as used in NeRF [31]. Since Spiking-Nerf [27] only reports PSNR, we present SSIM and LPIPS results for ANN-based NeRF, Spiking NeRF, and Spik-NeRF.

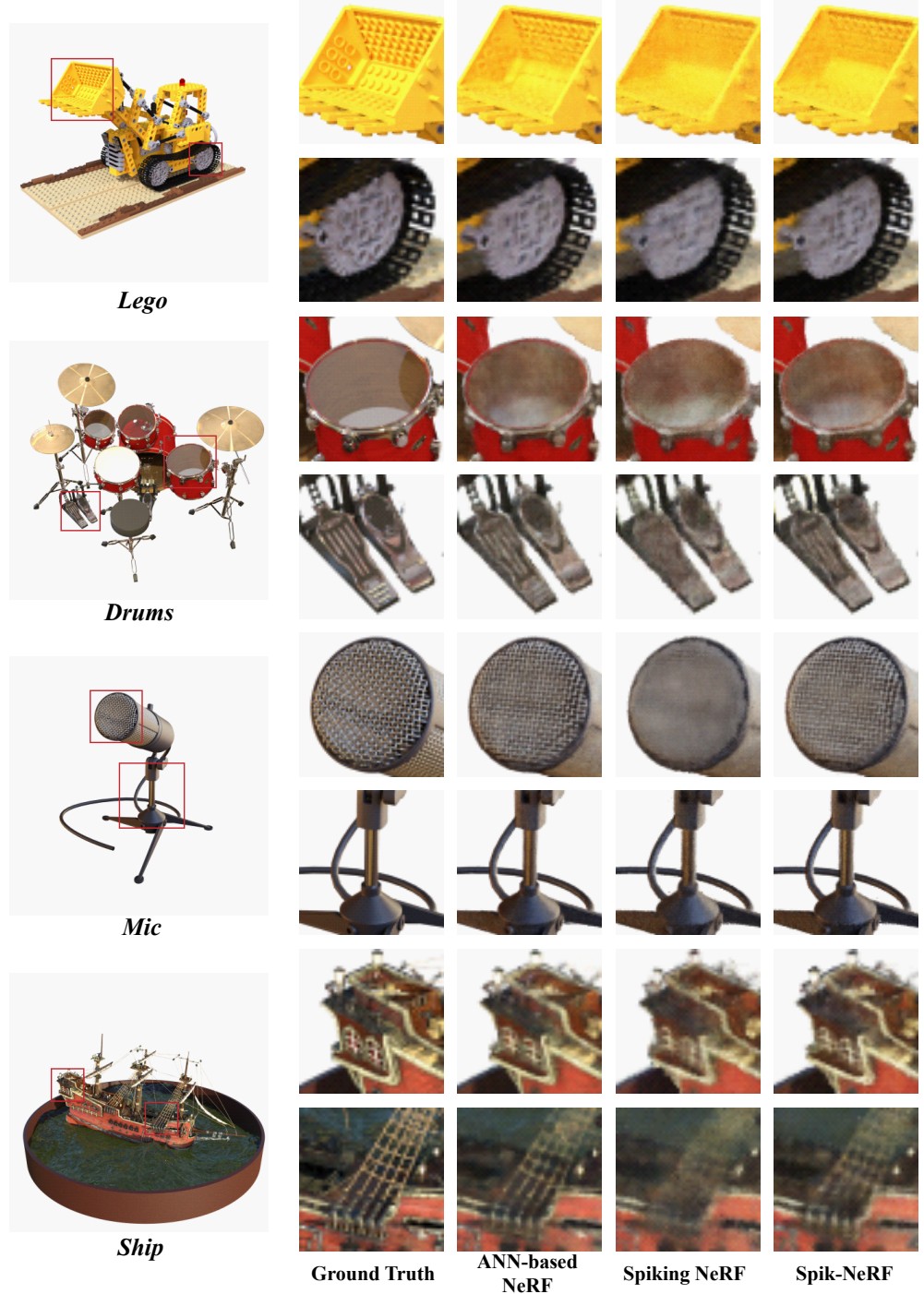

Figure 2: The rendering performance in the synthetic dataset.

On the synthetic dataset, Spiking-NeRF with 8 timesteps achieved an average PSNR of 11.93. In comparison, our directly trained Spiking NeRF achieved a significant improvement with an average PSNR of 28.95. More notably, our Spik-NeRF, utilizing ternary spikes, further boosts the average PSNR to 30.23, approaching the performance of the ANN-based NeRF (31.15 PSNR). Furthermore, in every scene, Spik-NeRF consistently outperforms both Spiking-NeRF and Spiking NeRF, highlighting the effectiveness of our approach.

On the realistic dataset, our method also surpasses Spiking-NeRF and Spiking NeRF. For instance, Spik-NeRF achieves a PSNR of 26.26, outperforming Spiking-NeRF and Spiking NeRF by 13.17 and 0.72 PSNR, respectively. In terms of SSIM and LPIPS, Spik-NeRF achieves scores of 0.817 and 0.121, while Spiking NeRF achieves 0.781 and 0.162, respectively.

Figure 2 illustrates the rendering results from synthetic datasets, including Lego, Drums, Mic, and Ship, for ANN-based NeRF, Spiking NeRF, and Spik-NeRF. It is evident that our method recovers fine details in both geometry and appearance, comparable to the ANN-based NeRF. This includes features such as the Drums' pedal, the Microphone's mesh grille, and the Ship's rigging. In contrast, Spiking NeRF produces blurry and distorted renderings, particularly for the Microphone's mesh grille.

We also include the rendering results for the realistic dataset in the appendix. As seen in the figures, our method consistently represents fine geometry more accurately across rendered views than Spiking NeRF.

## 4.2 Ablation Study for Knowledge Distillation

In this section, we evaluate the performance of our method with the proposed isomorphic network knowledge distillation on complex scenes, such as Room, Orchids, and Drums. The results are presented in Tab. 3.

As illustrated in Tab. 3, applying knowledge distillation to Spik-NeRF results in a noticeable improvement in performance, bringing its results much closer to those of the ANN-based NeRF. The quantitative analysis reveals two important findings: (1) Knowledge distillation effectively narrows the performance gap between Spik-NeRF and ANN-based NeRF, as demonstrated by the improvement in both PSNR and SSIM metrics. (2) Even in challenging scenes that involve specular reflections, such as Orchids, our method achieves rendering quality comparable to ANN-based NeRF, suggesting that knowledge distillation is beneficial in maintaining high-quality renderings in complex environments.

Table 3: Per-scene quantitative results for knowledge distillation.

| Metric | Method | Room | Orchids | Drums |
|---|---|---|---|---|
| PSNR↑ | ANN-based NeRF | 31.38 | 21.20 | 25.64 |
| | Spik-NeRF | 30.90 | 20.94 | 25.21 |
| | Spik-NeRF with KD | 31.12 | 21.09 | 25.40 |
| SSIM↑ | ANN-based NeRF | 0.931 | 0.734 | 0.929 |
| | Spik-NeRF | 0.920 | 0.688 | 0.921 |
| | Spik-NeRF with KD | 0.924 | 0.692 | 0.924 |
| LPIPS↓ | ANN-based NeRF | 0.049 | 0.122 | 0.053 |
| | Spik-NeRF | 0.066 | 0.170 | 0.065 |
| | Spik-NeRF with KD | 0.064 | 0.165 | 0.063 |

These findings also highlight the compatibility of our Spik-NeRF with the ternary spike neuron model and the isomorphic network knowledge distillation, which appears to facilitate the optimization process and enhance performance significantly.

## 5 Conclusion

We present Spik-NeRF, achieving ANN-comparable rendering quality. Theoretical analysis reveals binary spikes' limited representational capacity for SNN-based NeRF. To address limitations of binary spike neurons based Spiking NeRF, we propose the ternary spike neuron for Spik-NeRF, which increase representational capacity by 58.5% using three activation states. We also propose a train-infer decoupling via spike reparameterization technique to keep the energy efficiency of SNNs. In addition, we also propose the isomorphic distillation method, which transfers knowledge from ANN-based NeRF to compensate for information loss. Our experiments show that Spik-NeRF achieves PSNR metric within 2.9% of ANN baselines with only 2 timesteps, while retaining energy efficiency through multiplication-free operations. Our work bridges the efficiency-performance gap in neural fields, enabling future energy-efficient 3D reconstruction.

## Acknowledgements

This work was supported by the National Key Research and Development Program of China (No. 2024YDLN0013) and the National Natural Science Foundation of China (No. 12202412).

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

# A Technical Appendices and Supplementary Material

## A.1 Rendering Performance in Realistic Dataset

Figure 3 presents the rendering results from realistic datasets, including Flower, Room, T-rex, and Horns, for ANN-based NeRF, Spiking NeRF, and Spik-NeRF. As shown, our method captures fine details in both geometry and appearance, achieving results comparable to those of the ANN-based NeRF. In contrast, Spiking NeRF produces blurry and distorted renderings in certain areas.

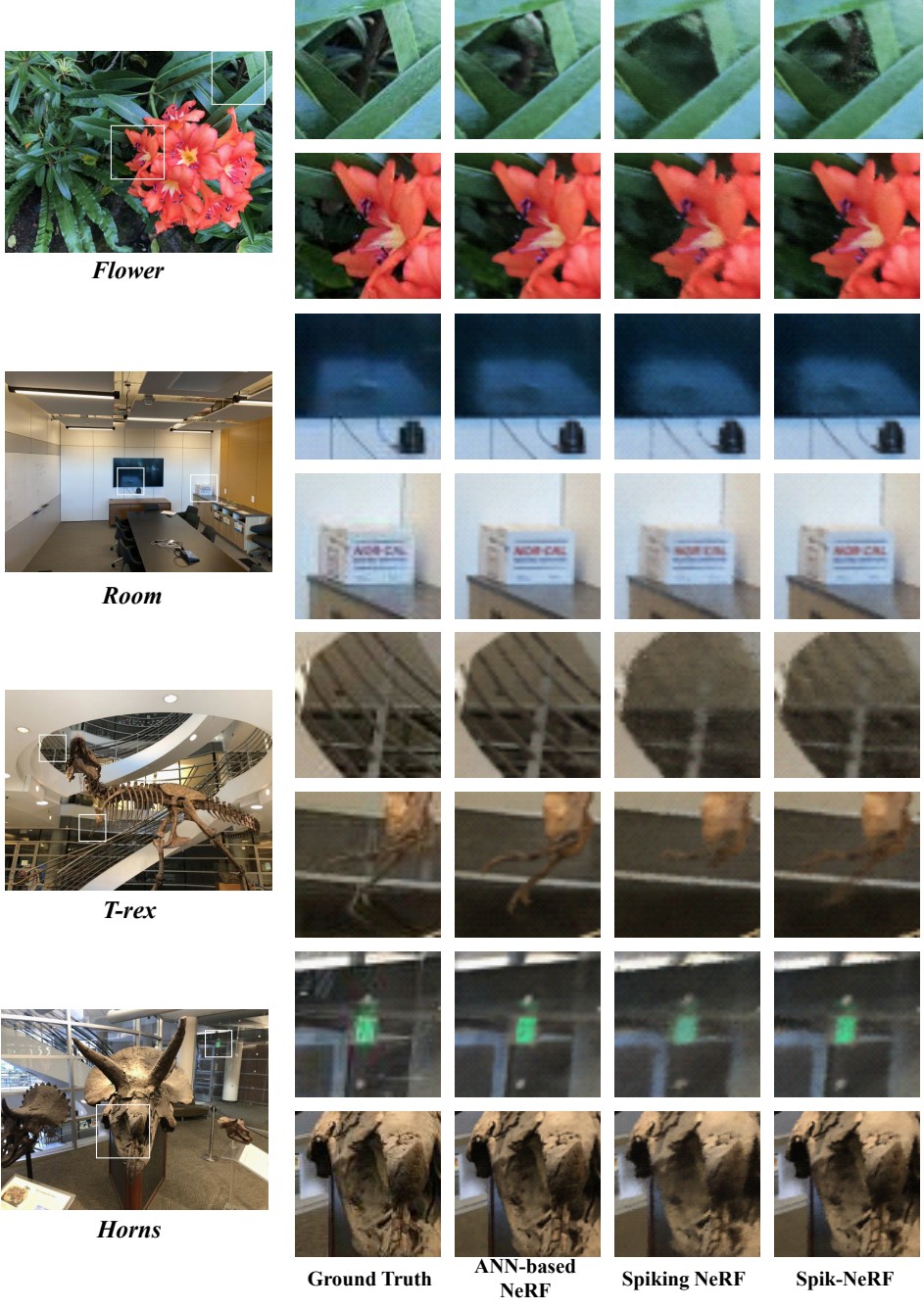

Figure 3: The rendering performance in the realistic dataset.

