# OpenReview forum: "Spik-NeRF: Spiking Neural Networks for Neural Radiance Fields"
_NeurIPS.cc/2025/Conference — NeurIPS 2025 poster_

### Official Review · Reviewer_iN4N · 2025-06-26

**Clarity:** 3
**Significance:** 3
**Originality:** 3
**Rating:** 5
**Confidence:** 5

**Summary:**

This paper explores the application of SNNs on the challenging rendering neural radiance fields (NeRF) tasks, and proposes the first directly trained SNN-based NeRF model called Spike-NeRF. To overcome the accuracy loss due to binary spikes, Spike-NeRF introduces ternary spiking neurons to enhance the model's information-carrying capacity. A reparameterization technique is further used to preserve efficient computation during inference. The authors additionally propose a distillation method to improve the performance of Spik-NeRF. Extensive experiments confirm the effectiveness and performance advantages of Spik-NeRF.

**Questions:**

1. Could the authors extend their experiments to include additional timesteps to better assess the temporal generalization of the proposed approach?

2. What is the rationale for training the proposed model for 300,000 iterations, especially given that the baseline NeRF model was trained for only 200,000? Could this discrepancy have influenced the reported performance gains?

**Ethical Concerns:**

["NO or VERY MINOR ethics concerns only"]

**Final Justification:**

The authors’ rebuttal has addressed my concerns about the generalization of Spike-NeRF to longer time steps and extended training iterations. In light of these clarifications, I am increasing my score and recommend acceptance.

**Limitations:**

yes

**Paper Formatting Concerns:**

No formatting issues.

**Quality:**

3

**Strengths And Weaknesses:**

Strengths:

1. The paper addresses the critical challenge of information loss in SNNs by introducing novel ternary spike neurons, which use activation values of 0/1/2 during training and −1/0/1 during inference by reparameterization. This design enhances the expressive capacity of SNNs while preserving their computational efficiency and event-driven nature.

2. The paper is well-written, with clear explanations of the methods, results, and conclusions.

3. The authors provide code and examples, which is a great help for reproducibility and a deeper understanding of the proposed methods.

Weaknesses:

1. All experiments were reported at a timestep of T=2. Since SNN behaviour often varies substantially with temporal resolution, it would be beneficial to evaluate the model using additional timesteps.

2. The comparison between models appears to be unfair. While the baseline NeRF model was trained for 200,000 iterations, the proposed model was trained for 300,000 iterations. This discrepancy in training duration could influence performance and may partially account for the observed improvements, rather than reflecting the superiority of the proposed method.

---

> ### Author Rebuttal · Authors · 2025-07-31
>
> Thanks for your efforts in reviewing our paper and your recognition of our novel method, clear presentation, and open source. The response to your questions is given piece by piece as follows.
>
> ---
>
> **W1**: All experiments were reported at a timestep of T=2. Since SNN behaviour often varies substantially with temporal resolution, it would be beneficial to evaluate the model using additional timesteps.
>
> **A1**: Thanks for the advice. We add the experiment results for 4 timesteps here. We did the experiments for Chair, Lego, and Horns, considering the limited time. It can be seen that the performance of Spik-NeRF with 4 timesteps could be improved further and is very close to the NeRF.
>
> |  Metric | Method | Chair | Lego | Horns |
> | --- | --- | --- | --- | --- |
> | PSNR | NeRF | 34.15 | 31.48 | 28.10 |
> |  | Spik-NeRF with 2 timesteps | 33.40 | 30.82 |  27.16 |
> |  | Spik-NeRF with 4 timesteps | 33.87 | 30.96 | 27.68 |
>
> ---
>
> **W2**: The comparison between models appears to be unfair. While the baseline NeRF model was trained for 200,000 iterations, the proposed model was trained for 300,000 iterations. This discrepancy in training duration could influence performance and may partially account for the observed improvements, rather than reflecting the superiority of the proposed method.
>
> **A2**: Thanks for the advice. Since SNN is more difficult to train, we use more training iterations as is commonly done in this field. Here, we also train the NeRF with 300,000 iterations for Chair, Lego, and Horns, considering the limited time. It can be seen that the performance remains almost the same as the iteration increases for the ANN-based NeRF. For a fair comparison, we choose their converged performance, and thus different iterations for ANN and SNN, respectively.
>
> |  Metric | Method | Chair | Lego | Horns |
> | --- | --- | --- | --- | --- |
> | PSNR | NeRF with 200,000 iterations | 34.15 | 31.48 | 28.10 |
> |  | NeRF with 300,000 iterations | 34.19 | 31.50 | 28.10 |
> |  | Spik-NeRF  | 33.40 | 30.82 |  27.16 |
>
> ---
>
> **Q1**: Could the authors extend their experiments to include additional timesteps to better assess the temporal generalization of the proposed approach?
>
> **A1**: Thanks for the question. Please see our response to **W1.**
>
> ---
>
> **Q2**: What is the rationale for training the proposed model for 300,000 iterations, especially given that the baseline NeRF model was trained for only 200,000? Could this discrepancy have influenced the reported performance gains?
>
> **A2**: Thanks for the question. Please see our response to **W2.**
>
> Thanks for your advice again. If you have any other questions, please feel free to let us know.

---

> > ### Comment · Reviewer_iN4N · 2025-08-02
> >
> > I appreciate the authors’ detailed response, which has resolved my concerns. Consequently, I am inclined to raise my score.

---

### Official Review · Reviewer_3BN5 · 2025-06-30

**Clarity:** 3
**Significance:** 3
**Originality:** 3
**Rating:** 5
**Confidence:** 4

**Summary:**

This paper explores the application of SNNs to the neural rendering task, specifically through the introduction of Spik-NeRF (Spiking Neural Radiance Fields with Ternary Spike). Traditional SNNs, which rely on binary spike activations, suffer from limited information capacity, leading to performance degradation in neural rendering models. To overcome this, the authors propose using ternary spike neurons, which significantly enhance the information-carrying capacity of the model. The result is a rendering performance in Spik-NeRF that is comparable to, or nearly identical with, traditional ANN-based NeRF models. Additionally, the authors introduce a re-parameterization technique to preserve the event-driven, multiplication-free advantages typical of binary spike neurons. Experimental results on both realistic and synthetic scenes show that Spik-NeRF achieves rendering performance comparable to ANN-based NeRF models.

**Questions:**

1. Could the authors extend their experimental evaluation to include other timestep values (e.g., 4) to assess how the ternary spike neurons perform across various temporal resolutions?

**Ethical Concerns:**

["NO or VERY MINOR ethics concerns only"]

**Final Justification:**

The authors have addressed  my concerns.  I am inclined to recommend acceptance.

**Limitations:**

See above.

**Quality:**

3

**Strengths And Weaknesses:**

Strengths:
1. This is one of the few works to apply SNNs to the NeRF task, potentially inspiring future research on low-power NeRF systems, which is a promising direction for energy-efficient rendering solutions.
2. The paper introduces an effective solution to the problem of information loss in traditional binary spike SNNs by utilizing ternary spike neurons. This enhancement increases the information-carrying capacity, significantly boosting performance in spiking neural rendering models.
3. The paper is well-written and clearly organized, making complex ideas accessible to a broader audience. The explanations of the methods and results are easy to follow.

Weaknesses:
1. The experiments conducted in the paper are only for a few timesteps. It would be good to test the proposed method with a broader range of timesteps to understand its performance across different configurations.
2. It does not compare its results to other works that have applied SNNs to NeRF. A comparison with existing studies in this area would help contextualize the significance and impact of this contribution.

---

> ### Author Rebuttal · Authors · 2025-07-31
>
> Thanks for your efforts in reviewing our paper and your recognition of our novel method and clear presentation. The response to your questions is given piece by piece as follows.
>
> ---
>
> **W1**: The experiments conducted in the paper are only for a few timesteps. It would be good to test the proposed method with a broader range of timesteps to understand its performance across different configurations.
>
> **A1**: Thanks for the advice. We add the experiment results for 4 timesteps here. We did the experiments for Chair, Lego, and Horns, considering the limited time. It can be seen that the performance of Spik-NeRF with 4 timesteps could be improved further and is very close to the NeRF.
>
> |  Metric | Method | Chair | Lego | Horns |
> | --- | --- | --- | --- | --- |
> | PSNR | NeRF | 34.15 | 31.48 | 28.10 |
> |  | Spik-NeRF with 2 timesteps | 33.40 | 30.82 |  27.16 |
> |  | Spik-NeRF with 4 timesteps | 33.87 | 30.96 | 27.68 |
>
> ---
>
> **W2**: It does not compare its results to other works that have applied SNNs to NeRF. A comparison with existing studies in this area would help contextualize the significance and impact of this contribution.
>
> **A2**: Thanks for the advice. There are 4 recent related SNN-NeRF variants. However, we feel only [1] is suitable to compare with us. In [1], an ANN-SNN model is proposed to develop energy-efficient spiking neural rendering using the ANN-to-SNN conversion approach, which requires many timesteps. We here report the detailed comparison below on both the synthetic dataset and the realistic dataset. It can be seen that ANN-SNN NeRF is much worse than our method. With only 2 timesteps, our method obtains similar performance to ANN-SNN NeRF, even with 256 timesteps.
>
> | PSNR | Synthetic dataset | Realistic dataset |
> | --- | --- | --- |
> | NeRF | 31.15 | 26.88 |
> | ANN-SNN NeRF with 8 timesteps | 11.93 | 13.09 |
> | ANN-SNN NeRF with 128 timesteps | 28.79 | 25.76 |
> | ANN-SNN NeRF with 256 timesteps | 30.41 | 26.31 |
> | Spik-NeRF with 2 timesteps | 30.23 | 26.26 |
>
> Other studies differ from our approach. For instance, hybrid ANN-SNN models were proposed in [2] and [3]. These works employ non-linear, non-spike functions to post-process the density-related outputs of the original ANN-based NeRF models. At the same time, these work did not report their PSNR. Thus, we can not compare our work with theirs. Another relevant study [4] presents a spiking neural radiance field model based on the DVGO and TensoRF frameworks, which are much different from our work. Additionally, it uses different datasets from our work. Thus, it is not suitable to compare our work with theirs either.
>
> [1] Spiking-NeRF: Spiking neural network for energy-efficient neural rendering
>
> [2] Spiking nerf: Representing the real-world geometry by a discontinuous representation
>
> [3] Sharpening Your Density Fields: Spiking Neuron Aided Fast Geometry Learning
>
> [4] SpikingNeRF: Making Bio-inspired Neural Networks See through the Real World
>
> ---
>
> **Q1**: Could the authors extend their experimental evaluation to include other timestep values (e.g., 4) to assess how the ternary spike neurons perform across various temporal resolutions?
>
> **A1**: Thanks for the question. Please see our response to **W1.**
>
> Thanks for your advice again. If you have any other questions, please feel free to let us know.

---

> > ### Comment · Reviewer_3BN5 · 2025-08-06
> >
> > I appreciate the authors’ detailed response. It has addressed my concerns, and I am inclined to keep my score.

---

### Official Review · Reviewer_E4R4 · 2025-07-02

**Clarity:** 3
**Significance:** 3
**Originality:** 4
**Rating:** 5
**Confidence:** 4

**Summary:**

This paper introduces Spik-NeRF, the first directly-trained Spiking Neural Network (SNN) approach for Neural Radiance Fields (NeRF). The authors identify the inherent limitation of binary spike activations in representing fine-grained scene information, which results in degraded rendering quality. To address this, they propose three technical contributions: 1) A ternary spike neuron to enhance representational capacity; 2) A re-parameterization scheme to convert ternary activations into a format amenable to efficient inference; 3) A distillation framework using a pre-trained ANN-based NeRF teacher to guide the SNN model. Extensive experiments on synthetic and real datasets demonstrate that Spik-NeRF approaches the rendering quality of ANN-based NeRF with significantly fewer timesteps, offering promise for energy-efficient 3D reconstruction.

**Questions:**

Why was only T=2 tested? How does Spik-NeRF quantitatively compare against other SNN-NeRF models?

**Ethical Concerns:**

["NO or VERY MINOR ethics concerns only"]

**Final Justification:**

Thank you for the detailed clarifications. I am inclined to recommend acceptance.

**Limitations:**

yes

**Quality:**

4

**Strengths And Weaknesses:**

Strengths:

1. Novelty and Contribution: This is the first fully SNN-based method directly trained on NeRF, which is a meaningful step in extending SNNs to high-dimensional neural rendering tasks.

2. Technical Insight: The paper provides solid theoretical justification for the use of ternary spike neurons, supported by entropy-based analysis. The proposed re-parameterization ensures energy-efficient inference without compromising the advantages of SNNs.

3. Distillation Strategy: The ANN-SNN distillation strategy is well-motivated and leads to performance improvements, especially in challenging scenes.

4. Clarity and Presentation: The paper is clearly written, well-organized, and easy to follow, with figures and tables that effectively support the claims.

Weaknesses:

1. The model is evaluated only at T=2. Since the energy-performance trade-off is central to SNN research, a more thorough analysis across different timesteps would be valuable.

2. While comparisons with Spiking-NeRF (Li et al., 2025) and ANN-based NeRF are provided, more detailed ablations against other recent SNN-NeRF variants would strengthen the empirical contributions.

3. The paper emphasizes energy efficiency, but no actual energy consumption or MAC operation estimates are reported. Including such results would better quantify the benefits of Spik-NeRF.

---

> ### Author Rebuttal · Authors · 2025-07-31
>
> Thanks for your efforts in reviewing our paper and your recognition of our well-motivated and novel method, solid theoretical justification, good results, and clear presentation. The response to your questions is given piece by piece as follows.
>
> ---
>
> **W1**: The model is evaluated only at T=2. Since the energy-performance trade-off is central to SNN research, a more thorough analysis across different timesteps would be valuable.
>
> **A1**: Thanks for the advice. We add the experiment results for 4 timesteps here. We did the experiments for Chair, Lego, and Horns, considering the limited time. It can be seen that the performance of Spik-NeRF with 4 timesteps could be improved further and is very close to the NeRF.
>
> |  Metric | Method | Chair | Lego | Horns |
> | --- | --- | --- | --- | --- |
> | PSNR | NeRF | 34.15 | 31.48 | 28.10 |
> |  | Spik-NeRF with 2 timesteps | 33.40 | 30.82 |  27.16 |
> |  | Spik-NeRF with 4 timesteps | 33.87 | 30.96 | 27.68 |
>
> ---
>
> **W2**: While comparisons with Spiking-NeRF (Li et al., 2025) and ANN-based NeRF are provided, more detailed ablations against other recent SNN-NeRF variants would strengthen the empirical contributions.
>
> **A2**: Thanks for the advice. There are 4 recent related SNN-NeRF variants. However, we feel only [1] is suitable to compare with us. In [1], an ANN-SNN model is proposed to develop energy-efficient spiking neural rendering using the ANN-to-SNN conversion approach, which requires many timesteps. We here report the detailed comparison below on both the synthetic dataset and the realistic dataset. It can be seen that ANN-SNN NeRF is much worse than our method. With only 2 timesteps, our method obtains similar performance to ANN-SNN NeRF, even with 256 timesteps.
>
> | PSNR | Synthetic dataset | Realistic dataset |
> | --- | --- | --- |
> | NeRF | 31.15 | 26.88 |
> | ANN-SNN NeRF with 8 timesteps | 11.93 | 13.09 |
> | ANN-SNN NeRF with 128 timesteps | 28.79 | 25.76 |
> | ANN-SNN NeRF with 256 timesteps | 30.41 | 26.31 |
> | Spik-NeRF with 2 timesteps | 30.23 | 26.26 |
>
> **Other studies differ from our approach.** For instance, hybrid ANN-SNN models were proposed in [2] and [3]. These works employ non-linear, non-spike functions to post-process the density-related outputs of the original ANN-based NeRF models. At the same time, these work did not report their PSNR. Thus, we can not compare our work with theirs. Another relevant study [4] presents a spiking neural radiance field model based on the DVGO and TensoRF frameworks, which are much different from our work. Additionally, it uses different datasets from our work. Thus, it is not suitable to compare our work with theirs either.
>
> [1] Spiking-NeRF: Spiking neural network for energy-efficient neural rendering
>
> [2] Spiking nerf: Representing the real-world geometry by a discontinuous representation
>
> [3] Sharpening Your Density Fields: Spiking Neuron Aided Fast Geometry Learning
>
> [4] SpikingNeRF: Making Bio-inspired Neural Networks See through the Real World
>
> ---
>
> **W3**: The paper emphasizes energy efficiency, but no actual energy consumption or MAC operation estimates are reported. Including such results would better quantify the benefits of Spik-NeRF.
>
> **A3**: Thanks for the advice.  Here, we add the energy comparison of NeRF and Spik-NeRF below. Following the practice in [Enabling Spike-Based Backpropagation for Training Deep Neural Network Architectures] and [Spiking-NeRF: Spiking Neural Network for Energy-Efficient Neural Rendering], we utilize 3.2pJ/MAC and 0.1pJ/AC as the energy consumption baseline. In NeRF, the number of MAC operations required for a single forward propagation process is 0.56M. In Spik-NeRF,  we first calculate the mean sparsity of activation of Spik-NeRF on the whole synthetic dataset using 2 timesteps, and it is 22.72%. Thus, the number of AC operations required for a single forward propagation process is 0.25M. It can be seen that the energy consumption of Spik-NeRF is much less than that of NeRF.
>
> |   | MACs | Energy  |
> | --- | --- | --- |
> | NeRF  | 0.56M | 1.792uJ |
> |  | ACs | Energy  |
> | Spik-NeRF(T=2) | 0.25M | 0.025uJ |
>
> ---
>
> **Q1**: Why was only T=2 tested? How does Spik-NeRF quantitatively compare against other SNN-NeRF models?
>
> **A1**: Thanks for the question. Please see our response to **W1 and W2.**
>
> Thanks for your advice again. If you have any other questions, please feel free to let us know.

---

### Official Review · Reviewer_dpTn · 2025-07-02

**Clarity:** 3
**Significance:** 2
**Originality:** 3
**Rating:** 4
**Confidence:** 3

**Summary:**

This paper introduces Spik-NeRF, a novel framework that adapts a fully SNN-based model for NeRF training. Through a rigorous theoretical analysis of the vanilla Spiking NeRF, the authors make several key contributions:
1.	Design of ternary spike neurons that significantly enhance the information-carrying capacity compared to standard binary activations.
2.	A reparameterization technique allowing ternary activations to be used efficiently in inference while retaining event-driven, multiplication-free advantages.
3.	An isomorphic network knowledge distillation scheme transferring knowledge from ANN-based NeRFs to Spik-NeRF.

**Questions:**

It is unclear whether the capacity can be further increased. For example, could quaternary spike neurons be used? Is there an upper bound on the number of discrete states that remain compatible with efficient SNN operation?

**Ethical Concerns:**

["NO or VERY MINOR ethics concerns only"]

**Final Justification:**

The authors address most of my concerns.

**Limitations:**

See above.

**Quality:**

2

**Strengths And Weaknesses:**

Strengths:
1.	The presentation of this paper is very clear and easy to understand.
2.	The designs of the ternary spike neurons and the reparameterization technique are well-motivated and appear effective.

Weaknesses:
1.	Although the authors claim in P3, L122, that the number of timesteps directly impacts the model’s energy efficiency, the experiments do not include any energy analysis. Compared to vanilla NeRF—which is now considered a relatively weak baseline—the approach may indeed consume less energy during inference, but the training overhead could be substantial due to the added complexity of ANN-NeRF distillation.
2.	The experiments do not report training time or inference time. Another concern is whether the proposed methods are compatible with faster NeRF variants such as InstantNGP, Nerfacto, or TensoRF. Focusing exclusively on vanilla NeRF may limit the practical relevance of the work for SNN applications in neural rendering or NVS.

---

> ### Author Rebuttal · Authors · 2025-07-31
>
> Thanks for your efforts in reviewing our paper and your recognition of our clear presentation, well-motivated and simple method, and solid theoretical analysis. The response to your questions is given piece by piece as follows.
>
> ---
>
> **W1**: Although the authors claim in P3, L122, that the number of timesteps directly impacts the model’s energy efficiency, the experiments do not include any energy analysis. Compared to vanilla NeRF—which is now considered a relatively weak baseline—the approach may indeed consume less energy during inference, but the training overhead could be substantial due to the added complexity of ANN-NeRF distillation.
>
> **A1**: Thanks for the advice. Here, **we add the energy comparison of NeRF and Spik-NeRF below**. Following the practice in [Enabling Spike-Based Backpropagation for Training Deep Neural Network Architectures] and [Spiking-NeRF: Spiking Neural Network for Energy-Efficient Neural Rendering], we utilize 3.2pJ/MAC and 0.1pJ/AC as the energy consumption baseline. In NeRF, the number of MAC operations required for a single forward propagation process is 0.56M. In Spik-NeRF,  we first calculate the mean sparsity of activation of Spik-NeRF on the whole synthetic dataset using 2 timesteps, and it is 22.72%. Thus, the number of AC operations required for a single forward propagation process is 0.25M. We also add the experiments with more timesteps for Spik-NeRF, When T = 4, the mean sparsity of activation of Spik-NeRF becomes 19.08% and the number of AC operations becomes 0.43M. **It can be seen that the energy consumption of Spik-NeRF is much less than that of NeRF. But as the timestep increases, the energy consumption of Spik-NeRF increases too.**
>
> |   | **MACs** | **Energy**  |
> | --- | --- | --- |
> | NeRF  | 0.56M | 1.792uJ |
> |  | **ACs** | **Energy**  |
> | Spik-NeRF(T=2) | 0.25M | 0.025uJ |
> | Spik-NeRF(T=4) | 0.43M | 0.043uJ |
>
> **For the training overhead, we agree that training overhead is a common concern in the SNN field. However, this is a challenge in this field, not only specific to our method.** Due to the inherent requirement of multiple timesteps in SNNs, their training typically incurs heavier burdens compared to ANNs, which is a well-recognized consensus in the community. To illustrate this, we have provided statistics on the training time for 100 iterations among NeRF, Spik-NeRF, and Spik-NeRF distilled from NeRF. These results clearly show that the training time of NeRF is significantly shorter than that of Spik-NeRF. **Even with ANN-NeRF distillation, our method only increases 16% training time compared to the baseline**. For the memory, it exhibits a similar phenomenon.
>
> |  Metric | Method | Chair | Lego | Horns |
> | --- | --- | --- | --- | --- |
> | Times | NeRF | 4.973s | 4.971s | 3.378s |
> |  | Spik-NeRF | 23.225s | 23.202s | 17.241s |
> |  | Spik-NeRF distilled from NeRF | 27.701s | 27.672s | 20.661s |
> | Memory | NeRF | 5435MB | 5435MB | 5009MB |
> |  | Spik-NeRF | 22469MB | 22469MB | 17937MB |
> |  | Spik-NeRF distilled from NeRF | 25423MB | 25421MB | 20185MB |
>
> While we acknowledge that our approach does have a higher training burden compared to vanilla ANNs, **we would like to emphasize its superiority over existing spiking NeRF methods in terms of training efficiency. Most current spiking NeRF methods require at least 10 timesteps to achieve satisfactory performance. In contrast, our method only needs 2 timesteps.** We have also included the training burden of Spiking NeRF with 10 timesteps for comparison, which clearly demonstrates the substantial advantage of our approach in reducing training overhead. Thus, our method effectively alleviates the training burden commonly associated with SNNs, making it more practical for real-world applications.
>
> |  Metric | Method | Chair | Lego | Horns |
> | --- | --- | --- | --- | --- |
> | Times | Spik-NeRF with 2 timesteps | 23.225s | 23.202s | 17.241s |
> |  | Spiking NeRF with 10 timesteps | 116.265s | 115.991s | 86.220s |
>
> ---
>
> **W2**: The experiments do not report training time or inference time. Another concern is whether the proposed methods are compatible with faster NeRF variants such as InstantNGP, Nerfacto, or TensoRF. Focusing exclusively on vanilla NeRF may limit the practical relevance of the work for SNN applications in neural rendering or NVS.
>
> **A2**: Thanks for the advice. Here, we add statistics on the training time for 100 iterations and inference time for one image of Chair, Lego, and Horns, among NeRF, Spik-NeRF, and Spik-NeRF distilled from NeRF. We will add this to the final paper version. **It should be noted that the training time and inference time of SNNs are worse than ANNs. This is an inherent problem in the field of SNN.**
>
> |  Metric | Method | Chair | Lego | Horns |
> | --- | --- | --- | --- | --- |
> | Training times for 100 iterations | NeRF | 4.973s | 4.971s | 3.378s |
> |  | Spik-NeRF | 23.225s | 23.202s | 17.241s |
> |  | Spik-NeRF distilled from NeRF | 27.701s | 27.672s | 20.661s |
> | Inference time for one image | NeRF | 2.405s | 2.370s | 2.100s |
> |  | Spik-NeRF | 8.215s | 8.375s | 7.335s |
> |  | Spik-NeRF distilled from NeRF | 8.211s | 8.302s | 7.296s |
>
> In addition, our method is also compatible with faster NeRF variants. We would like to address it from two aspects:
>
> First, the core of Spik-NeRF lies in improving the neural network components within NeRF—specifically, optimizing the feedforward neural networks used for predicting properties like density and color. This design philosophy is inherently transferable to other NeRF variants, since all these variants fundamentally rely on neural networks to model and predict scene-related information. Thus, the idea of enhancing neural network components with spiking mechanisms, as in Spik-NeRF, can be naturally extended to these variants.
>
> Second, to validate this transferability, we have applied Spik-NeRF’s core idea to TensoRF with 2 timesteps for the TensoRF-CP-384 backbone. We did the experiments for Chair and Lego, considering the limited time. **The results demonstrate that our method retains its effectiveness in this setting.**
>
> |  Metric | Method | Chair | Lego |
> | --- | --- | --- | --- |
> | PSNR | TensoRF | 33.41 | 31.23 |
> |  | Spik-TensoRF | 32.98 | 30.62 |
>
> ---
>
> **Q1**: It is unclear whether the capacity can be further increased. For example, could quaternary spike neurons be used? Is there an upper bound on the number of discrete states that remain compatible with efficient SNN operation?
>
> **A1**: Thanks for the advice. **The capacity can be further increased.** We add the experiment results for quaternary spike neurons here. We did the experiments for Chair and Lego, considering the limited time. It can be seen that the performance of Spik-NeRF with quaternary spike is almost the same as NeRF. **However, to retain the event-driven, multiplication-free advantages typical of SNNs, 3 is the upper bound on the number of discrete states**, since only {-1,0,1} could be chosen for spike to keep multiplication-free advantages.
>
> |  Metric | Method | Chair | Lego |
> | --- | --- | --- | --- |
> | PSNR | NeRF | 34.15 | 31.48 |
> |  | Spik-NeRF with ternary spike | 33.40 | 30.82 |
> |  | Spik-NeRF with quaternary spike | 34.08 | 31.35 |
>
> Thanks for your advice again. If you have any other questions, please feel free to let us know.

---

> > ### Comment · Reviewer_dpTn · 2025-08-02
> >
> > Thanks for the authors' response. It addresses most of my concerns. I will raise my score to 4.

---

### Decision · Program_Chairs · 2025-09-17

**Decision:**

Accept (poster)

**Comment:**

This paper received uniformly positive reviews. The work presents promising results on energy-efficient spiking neural networks (SNNs) for neural rendering applications. In particular, the use of ternary spike neurons is a notable strength, and the reviewers also appreciated the clarity and quality of the paper’s presentation.

Overall, this is a good paper whose technical contribution and clear presentation make it valuable to the community, and I support its acceptance.